# Scalable production of structurally colored composite films by shearing supramolecular composites of polymers and colloids

Miaomiao Li[1], Bolun Peng[1], Quanqian Lyu[1], Xiaodong Chen[1], Zhen Hu[1], Xiujuan Zhang[1], Bijin Xiong[1], Lianbin Zhang [1] ✉ & Jintao Zhu [1]

Structurally colored composite films, composed of orderly arranged colloids in polymeric matrix, are emerging flexible optical materials, but their production is bottlenecked by time-consuming procedures and limited material choices. Here, we present a mild approach to producing large-scale structurally colored composite films by shearing supramolecular composites composed of polymers and colloids with supramolecular interactions. Leveraging dynamic connection and dissociation of supramolecular interactions, shearing force stretches the polymer chains and drags colloids to migrate directionally within the polymeric matrix with reduced viscous resistance. We show that meter-scale structurally colored composite films with iridescence color can be produced within several minutes at room temperature. Significantly, the tunability and diversity of supramolecular interactions allow this shearing approach extendable to various commonly-used polymers. This study overcomes the traditional material limitations of manufacturing structurally colored composite films by shearing method and opens an avenue for mildly producing ordered composites with commonly-available materials via supramolecular strategies.

With the increasing demand for a sustainable society, growing efforts are currently devoted to developing more environmentally friendly and economic approaches to produce functional nanomaterials with visible color as a promising alternative to conventional synthetic dyes[1–10]. Among the various functional nanomaterials, structurally colored composite films (SCCFs) composed of orderly arranged colloids embedded in the polymeric matrix have gained immense interest in scientific and industrial communities[11,12]. These SCCFs process both the structural color endowed by the orderly arrangement of colloids and the flexibility and functionality of polymers[1,8,11,13], making them promising for applications in various fields, including displays[14–16], anti-counterfeiting[17,18], colorful coatings[19,20], visual sensing[21–23], solar cells[24], and optical fibers[25].

Generally, solvent evaporation-induced colloidal assembly is commonly used to produce SCCFs from colloids[26–28]. This assembly process is conducted in a liquid dispersion medium, where the ordered arrangement of colloids is driven by the dynamic balance between electrostatic repulsion, solvation forces, van der Waals forces, and capillary forces generated by the volatilization of the medium[19,29–31]. This process, however, requires elegant control of translational colloidal diffusion, crystal nucleation, and crystal growth (usually taking hours to days)[29,32,33]. Thus, it is difficult to obtain an ordered arrangement of colloids over a large area in a short period (with areas usually limited to a few square centimeters). Moreover, introducing additional polymers into ordered colloidal arrangements further adds to the difficulty of producing SCCFs[12,28]. Therefore, rapid and large-scale production of SCCFs remains challenging.

Different from solvent evaporation-induced colloidal assembly, oriented external forces, such as electrical[34,35], magnetic[3,36], and shear flow forces[37,38], have an improved capability to rapidly induce the

[1]State Key Laboratory of Material Processing and Die & Mould Technology and School of Chemistry and Chemical Engineering, Huazhong University of Science and Technology (HUST), Wuhan 430074, China. ✉e-mail: zhanglianbin@hust.edu.cn

ordered arrangement of colloids (usually within a few minutes) by overcoming their thermal motion and are thus promising for the rapid fabrication of SCCFs[37,39]. In particular, shearing forces can provide momentum for the directional migration of colloids in a dispersive viscoelastic medium to arrange over macroscopic length scales[39,40]. Although such a shearing technique can achieve rapid and large-scale manufacturing of SCCFs, it is unfortunately limited to colloids with core-interlayer-shell structures, in which polymers with a low glass transition temperature ($T_g$, usually below room temperature) serve as the shell and are covalently grafted onto the surface of a rigid core by an interlayer[38,39,41–43]. In such a system, shearing forces drag the rigid core colloid to migrate into an ordered arrangement by stretching the covalently-linked shell polymer chains and colloidal collisions[42,44]. To improve the processing performance, it is often necessary to melt the polymer shell matrix at elevated temperatures to decrease the viscous resistance of colloidal migration[37,38,45], which unfortunately necessitates significant energy consumption during processing. Therefore, developing universal and mild shear-induced colloidal ordering techniques suitable to commonly-used polymers and colloids is imperative to advance the development of SCCFs but remains challenging.

Supramolecular interactions with broad tunability and diversity have shown great potential in conferring many fascinating properties to functional materials[46–49]. Notably, due to their inherent dynamic and reversible nature, supramolecular interactions can dissociate and reconstruct autonomously or in response to external stimuli such as stress, solvents, pH, and temperature, allowing the resultant materials to exhibit shear thinning properties and to be efficiently processed under mild conditions[16,47,48,50]. Thus, we hypothesized that by leveraging optimized supramolecular interactions between polymers and colloids, shearing techniques could be effective for constructing SCCFs in a rapid, mild, and scalable manner. On the one hand, the supramolecular interactions, like covalent bonds in the core-interlayer-shell colloids, can readily connect colloids and polymers, effectively transferring momentum between the polymeric matrix and colloids to obtain an ordered colloidal arrangement. On the other hand, supramolecular interactions, different from covalent links, are diverse and dynamically reversible, offering the possibility of mild and scalable production of SCCFs with high-quality structural colors using a wide range of available materials.

Here, as a proof of concept, we construct supramolecular composites composed of colloids and polymers and subject them to shearing forces. Specifically, we employ carboxylated polystyrene (PS-COOH) colloids and polyethyleneimine (PEI) to obtain supramolecular composites based on hydrogen bonding and electrostatic interactions. By shearing these composites on a customized roll-to-roll setup, a 4-meter-long SCCF with iridescent color can be produced within three minutes. The dynamic nature of supramolecular interactions significantly improved the processing performance of the composite, enabling the production of SCCFs at milder conditions. Notably, the colloidal ordering mechanism during shearing supramolecular composites is elucidated, providing guidance and insights for designing general formulations for producing SCCFs with brilliant structural colors. This approach can be expanded to various commonly-used polymers, demonstrating excellent generality and versatility. This study provides insight into shear-induced colloidal ordering and paves the way for mildly constructing SCCFs using a supramolecular strategy.

## Results

### Construction of SCCFs by shearing supramolecular composites

The key to the mild and scalable production of high-quality SCCFs via shearing technology with commonly-used polymers and colloids lies in the rational introduction of supramolecular composites. Via optimized supramolecular interactions, colloids and polymers can be effectively connected while dynamically dissociated, allowing for comfortably transferring momentum between the polymeric matrix and colloids to

obtain an ordered colloidal arrangement under shearing. Taking PS-COOH colloids and PEI polymer as an example, we show the preparation and optimization of supramolecular composites. A series of uniformly sized PS-COOH colloids with different diameters (from 177 to 225 nm) were synthesized using a soap-free emulsion method (Supplementary Fig. 1). Subsequently, the supramolecular composites were readily obtained by physically mixing an aqueous PS-COOH dispersion with a PEI solution and evaporating the solvent, resulting in solid-like composites, denoted as $PEI_x$-$PS_y$ (Fig. 1a(i)). In this notation, $x$ represents the value corresponding to the number-average molecular weight ($M_n$) of PEI, and $y$ represents the volume fraction of PS-COOH ($\varphi_{PS}$). Analysis using Fourier transform infrared (FTIR) spectroscopy and zeta potential measurements confirmed the adsorption of PEI onto the surface of PS-COOH through supramolecular hydrogen bonding and electrostatic interactions between the amino groups of PEI and the carboxyl groups of PS-COOH[51] (Supplementary Fig. 2 and Supplementary Note 1). Notably, this supramolecular adsorption can bring about additional chain entanglement, known as adsorption-induced entanglements[52] (Fig. 1a). These entanglements effectively connect the colloids with the polymeric matrix in the composite and potentially contribute to momentum transfer under shearing. As a result, the PS-COOH colloids were uniformly dispersed in the polymeric matrix, as evidenced by the discrete circular ring pattern in the two-dimensional (2D) fast Fourier transform (FFT) image and scanning electron microscopy (SEM) observation[33] (Supplementary Fig. 3). By comparing the dispersion and the corresponding shear-induced ordering of colloids in composites constructed from colloids with different surface charges and PEIs, we found that the uniform distribution of colloids within the polymeric matrix is a prerequisite for shear-induced colloidal ordering (Supplementary Fig. 4 and Supplementary Note 2). Therefore, taking the $PEI_{60k}$-$PS_{60}$ composite as an example, we show the process for constructing SCCFs by shearing such supramolecular composites.

In this study, we opted for bending-induced shearing treatment (BIST) to subject supramolecular composites to controlled shear, which allowed us to study the mechanism of shear-induced colloidal ordering during SCCF formation and thus develop a general-purpose model and couple it with a controlled roll-to-roll process for continuous production at scale[37,39]. Specifically, the as-prepared supramolecular composite was first sandwiched between two polyethylene terephthalate (PET) sheets (thickness: 57 μm) and laminated into a composite film with a sandwich structure under pressure (Fig. 1a(ii)). After lamination, the sandwich film was bent along a fixed rod and reciprocated under a specific load (Fig. 1a(iii)). Since PET is far more rigid than the composite, bending this sandwich film generated a strong shearing force inside the composite film parallel to the surface[37]. Under the shearing force, the PEI chains slipped and stretched, partially dissociating supramolecular interactions between the polymer and colloids[53] (Fig. 1b(i)). As a result, the colloids could be dragged to migrate directionally within the matrix with reduced viscous resistance, corresponding to an increase in the ordering of the colloidal arrangement. Rheological measurements revealed that the viscosity of the $PEI_{60k}$-$PS_{60}$ composite decreased by ~2–3 orders of magnitude across oscillation frequencies ranging from 0.1 to 100 Hz (Supplementary Fig. 5). This means that the viscous resistance of colloidal migration is significantly reduced under shearing, which is advantageous for transforming the colloidal arrangement. Upon removal of the shearing force, the dissociated supramolecular interactions are reconstructed autonomously, and the resulting ordered colloidal arrangement can be fixed within the polymeric matrix with the resumed viscosity (Fig. 1b(ii)). After several reciprocal shearing processes (i.e., shearing passes), an SCCF with a highly ordered colloidal arrangement and bright color was produced. To showcase the large-scale production of SCCFs, a roll-to-roll processing setup was customized, which includes two rolling pressure rollers with a diameter of 10 cm and five fixed rods with a diameter of 6 mm (Fig. 1c).

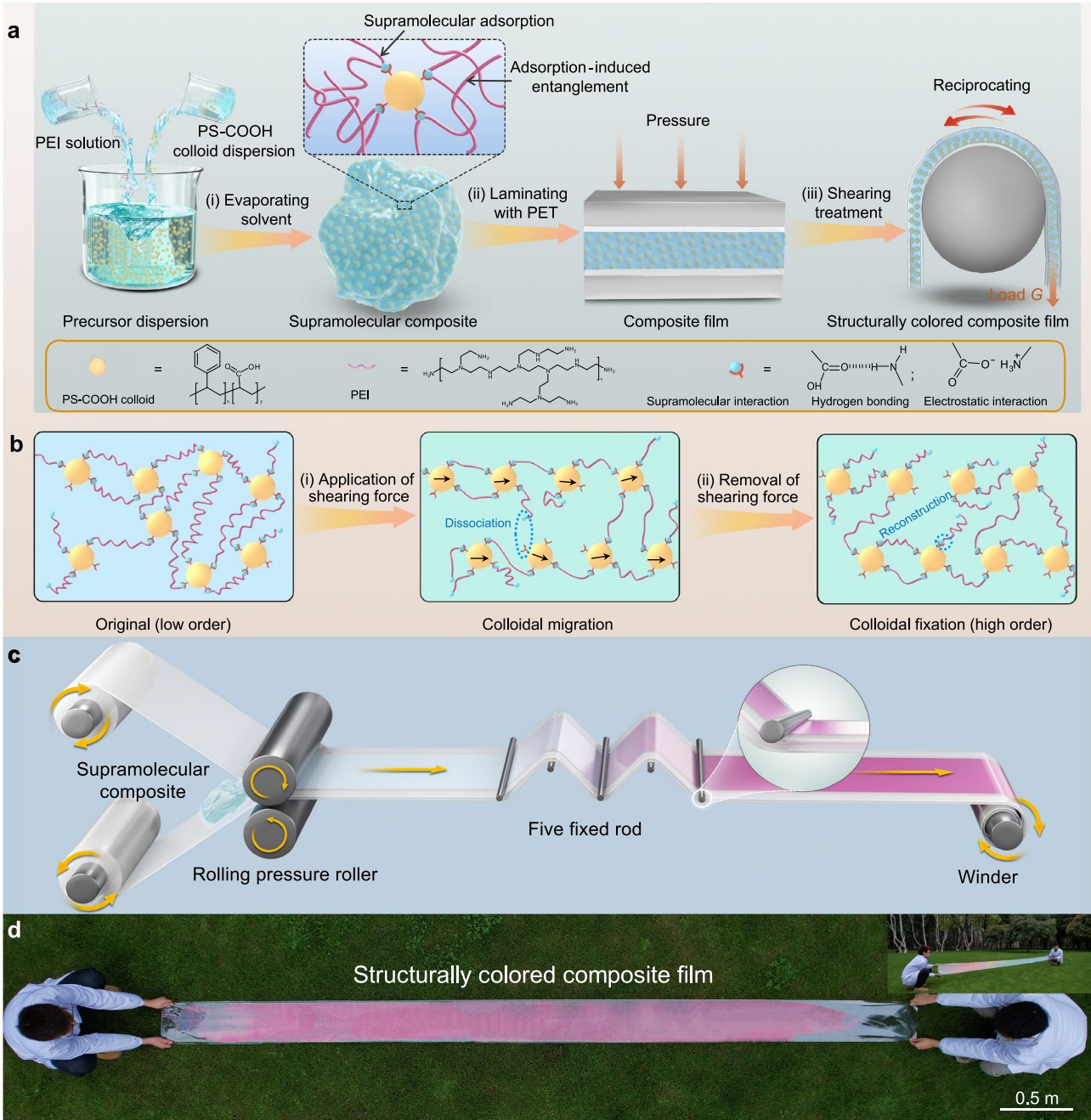

**Fig. 1 | Construction of SCCFs by shearing supramolecular composites.**
**a** Scheme for the preparation, lamination with polyethylene terephthalate (PET) sheets, and shearing treatment of the supramolecular composite. The enlarged image shows the interaction between carboxylated polystyrene (PS-COOH) colloids and polyethyleneimine (PEI), including supramolecular adsorption and adsorption-induced chain entanglement. **b** Scheme showing colloidal migration and fixation during shearing treatment of the supramolecular composite. **c** Scheme of the production of SCCFs using a customized continuous roll-to-roll setup. **d** SCCF obtained through a roll-to-roll production process. Inset in (**d**): photograph of the twisted SCCF (sample size: $0.3 \times 4.3$ m$^2$).

These rods ensured the generation of significant shearing forces within the supramolecular composites during processing, facilitating the production of SCCFs with bright colors. Following the roll-to-roll processing, a four-meter-long SCCF with red color was obtained within a three-minute timeframe (Fig. 1d). When the film was twisted, it exhibited iridescence due to the strong angle dependence of its structural color.

## Optimization of supramolecular composites

Shear-induced colloidal ordering in the supramolecular composites depends on the colloids' ability to acquire momentum during shearing[44], which involves energy dissipation and transfer and is highly correlated with the composition of the supramolecular composite[23,54]. Therefore, careful optimization of the composite composition and clarification of the mechanism of shear-induced colloidal ordering is essential, which can provide crucial guidance for designing composite formulations for producing SCCFs with high-quality structural colors. To compare the ordering effects produced by shearing different supramolecular composites, optimizing the shear processing parameters is a prerequisite. The optimization process was first implemented on a laboratory-scale fixed rod, as depicted in Fig. 1a(iii). Since the shear-induced ordering effect of the colloids in the composite

depends on the applied shear strain ($\gamma$)[37], the shear strain rate ($\dot{\gamma}$)[23], and the resulting shear stress and compressive stress[55] in the composite, the relevant processing parameters, including the thickness of the supramolecular composite ($h_{Composite}$), the diameter of the rod ($D$), and the load ($G$), were optimized individually while other parameters were held constant. The calculations of $\gamma$ (Supplementary Note 3), $\dot{\gamma}$ (Supplementary Note 4), average compressive stress (Supplementary Note 5), shear stress (Supplementary Note 6), as well as the corresponding optimization results (Supplementary Figs. 6–9) are described in the Supplementary Information. As a result, the optimized processing parameters were determined to be $h_{Composite} = \sim 100$ μm, $D = 3$ mm, and $G = 5$ N. It is essential to note that, unlike previous colloidal systems with core-interlayer-shell structures, we found that the ordering effect of the current supramolecular composite was strongly influenced by the compressive stress applied by fixed rods. Therefore, fixed rods with relatively smaller diameters (i.e., ~3–8 mm) are more suitable for achieving the desired ordering effect.

After optimizing the processing parameters, the composition of the supramolecular composite, including the polymer molecular weight and the colloidal volume fraction, was carefully investigated. To investigate the role of polymer molecular weight on shear-induced colloidal ordering, different PEI with $M_n$ values of 600, 10k, and 60k (denoted $PEI_{600}$, $PEI_{10k}$, and $PEI_{60k}$, respectively) were used to prepare supramolecular composites. To compare the effect of polymer molecular weight on shear-induced ordering, we subjected the composite films to sufficient shearing treatments, which referred to the composite film with the highest reflectivity observed in their reflection spectra during shearing[37]. After sufficient shearing treatment, the composite films constructed from these PEIs showed distinct appearances. The $PEI_{600}$-$PS_{60}$ film showed no noticeable changes before and after shearing treatments, and the films appeared blue with angle-independent structural colors (Fig. 2a(i)). This result was consistent with the low order of the internal colloidal arrangement revealed by discrete circular ring patterns in the 2D FFT images and SEM observations[31] (Fig. 2a(i)). Additionally, almost no peaks were observed in the reflection spectra before and after shearing treatments (left panel of Fig. 2b), indicating the ineffectiveness of shear-induced ordering of colloids in the $PEI_{600}$-$PS_{60}$ composite on the macroscale[33]. In contrast, for the $PEI_{10k}$-$PS_{60}$ and $PEI_{60k}$-$PS_{60}$ films, significant changes in appearance were observed before and after sufficient shearing treatments. Before shearing treatment, these composite films exhibited blue, angle-independent structural colors and low-ordered colloidal arrangements (top panels of Fig. 2a(ii) and (iii)), similar to the $PEI_{600}$-$PS_{60}$ composite. After sufficient shearing treatments, both composite films showed angle-dependent structural colors, with the color changing from cyan to blue as the viewing angle varied from 0° to 60° (bottom panels of Fig. 2a(ii) and (iii)). Moreover, the 2D FFT images of sheared $PEI_{10k}$-$PS_{60}$ and $PEI_{60k}$-$PS_{60}$ films transformed from discrete circular ring patterns to sharp hexagonal peak patterns (bottom panels of Fig. 2a(ii) and (iii)), demonstrating the generation of highly ordered structures in these films after shearing treatments[33]. Accordingly, compared to the $PEI_{60}$-$PS_{60}$ film, the sufficiently sheared $PEI_{10k}$-$PS_{60}$ and $PEI_{60k}$-$PS_{60}$ films exhibited higher peak intensities (~50% and 52%, respectively) in the reflection spectra (middle and right panels of Fig. 2b). These results suggest a strong correlation between the shear-induced colloidal ordering effect and the polymer molecular weight in the supramolecular composite. This intriguing discovery inspired our subsequent exploration of the ontological properties of these composites.

To elucidate the mechanism of shear-induced colloidal ordering in supramolecular composites, the viscoelastic properties of these PEIs and their corresponding composites were analyzed through rheological measurements, specifically the frequency dependence of the shear moduli (i.e., storage modulus ($G'$) and loss modulus ($G''$)). It should be noted that for $PEI_{600}$, its viscosity was independent of frequency (Supplementary Fig. 10), implying that $PEI_{600}$ behaved like a Newtonian fluid with negligible elasticity[56]. For $PEI_{10k}$, its $G''$ was proportional to $\omega^2$, and G' was proportional to $\omega$ (Fig. 2c), indicating that $PEI_{10k}$ was a typical nonentangled polymer[56]. In contrast, the G' and G" of $PEI_{60k}$ were highly dependent on frequency, implying that $PEI_{60k}$ was a typical viscoelastic liquid caused by chain entanglement[56]. In comparison, in the case of the $PEI_{600}$-$PS_{60}$ composite, a crossover of the dynamic shear modulus was observed in the low-frequency region (orange lines of Fig. 2d), indicating the occurrence of terminal flow[52] at angular velocities above 20 rad s$^{-1}$, corresponding to $\dot{\gamma} \sim 250$ s$^{-1}$. This $\dot{\gamma}$ value closely matches the shear rate during shearing treatment of the supramolecular composite (278 s$^{-1}$, calculated in the Supplementary Note 4), suggesting that $PEI_{600}$ in the $PEI_{600}$-$PS_{60}$ composite undergoes whole chain relaxation[57]. These results imply that the $PEI_{600}$ in the $PEI_{600}$-$PS_{60}$ composite failed to be stretched during shearing, and the corresponding energy was dissipated completely through the rapid relaxation of the polymer chains[23] (Fig. 2e(i)). Therefore, it can be speculated that the colloids in this composite gain negligible momentum, failing to yield an ordered colloidal arrangement (Fig. 2e(ii)).

In sharp contrast, the dynamic shear moduli of the $PEI_{10k}$-$PS_{60}$ and $PEI_{60k}$-$PS_{60}$ composites were found to be parallel rather than crossover (green and blue lines of Fig. 2d), indicating that PEI in these composites underwent segmental chain relaxation instead of whole chain relaxation[52]. Thus, these composites were more elastic over the studied frequency ranges than $PEI_{600}$-$PS_{60}$. The increase in elasticity can be attributed to the topological constraint of the polymer chains by supramolecular adsorption and adsorption-induced chain entanglement brought about by the high molecular weight polymer[52], which reduces chain slippage and increases polymer chain stretching. Under constant applied stress, the time-dependent strain curves showed that the strain rate of the composite decreased with increasing polymer molecular weight (Supplementary Fig. 11), which supported the reduction in chain slippage caused by the increase in molecular weight. These results imply that under the current shearing conditions, the energy generated in the $PEI_{10k}$-$PS_{60}$ and $PEI_{60k}$-$PS_{60}$ composites was not completely dissipated through segmental chain relaxation but rather partially converted into elastic energy through chain stretching[23] (Fig. 2e(iii)). Therefore, the colloids in the $PEI_{10k}$-$PS_{60}$ and $PEI_{60k}$-$PS_{60}$ composites could gain momentum through polymer stretching, and colloidal collisions occurred during shearing, resulting in an ordered colloidal arrangement[44] (Fig. 2e(iv)). Interestingly, we found that the shear-induced colloidal ordering in the supramolecular composites was also applicable to other polymers (e.g., poly(ethylene glycol)-*block*-poly(propylene glycol)-*block*-poly(ethylene glycol), PEG-*b*-PPG-*b*-PEG), and the ordering effect was also associated with their $M_n$ (Supplementary Fig. 12 and Supplementary Note 7). These results elucidate a universal mechanism whereby chain entanglement and topological constraints brought about by high molecular weight polymers in supramolecular composites can reduce the energy dissipation of the polymeric matrix and promote the colloid to gain momentum during shearing, ultimately resulting in ordered colloidal arrangements. It is noteworthy that the influence of polymer molecular weight on the shear-induced ordering effect does not show a strict positive correlation. When the PEI molecular weight increased from 60 to 2000 kDa, the peak intensity of the sufficient sheared SCCF decreased from 55% to 4% (Supplementary Fig. 13). This indicates that the shear-induced ordering effect significantly diminished with further increase in PEI molecular weight. This phenomenon may be attributed to the tendency of ultra-high molecular weight PEIs to undergo excessive entanglement, which enhances interactions between polymer matrices, increases energy dissipation, and ultimately hinders effective momentum transfer.

To explore another factor affecting shear-induced colloidal ordering in supramolecular composites, i.e., the volume fraction of colloids, we constructed a series of supramolecular composites with

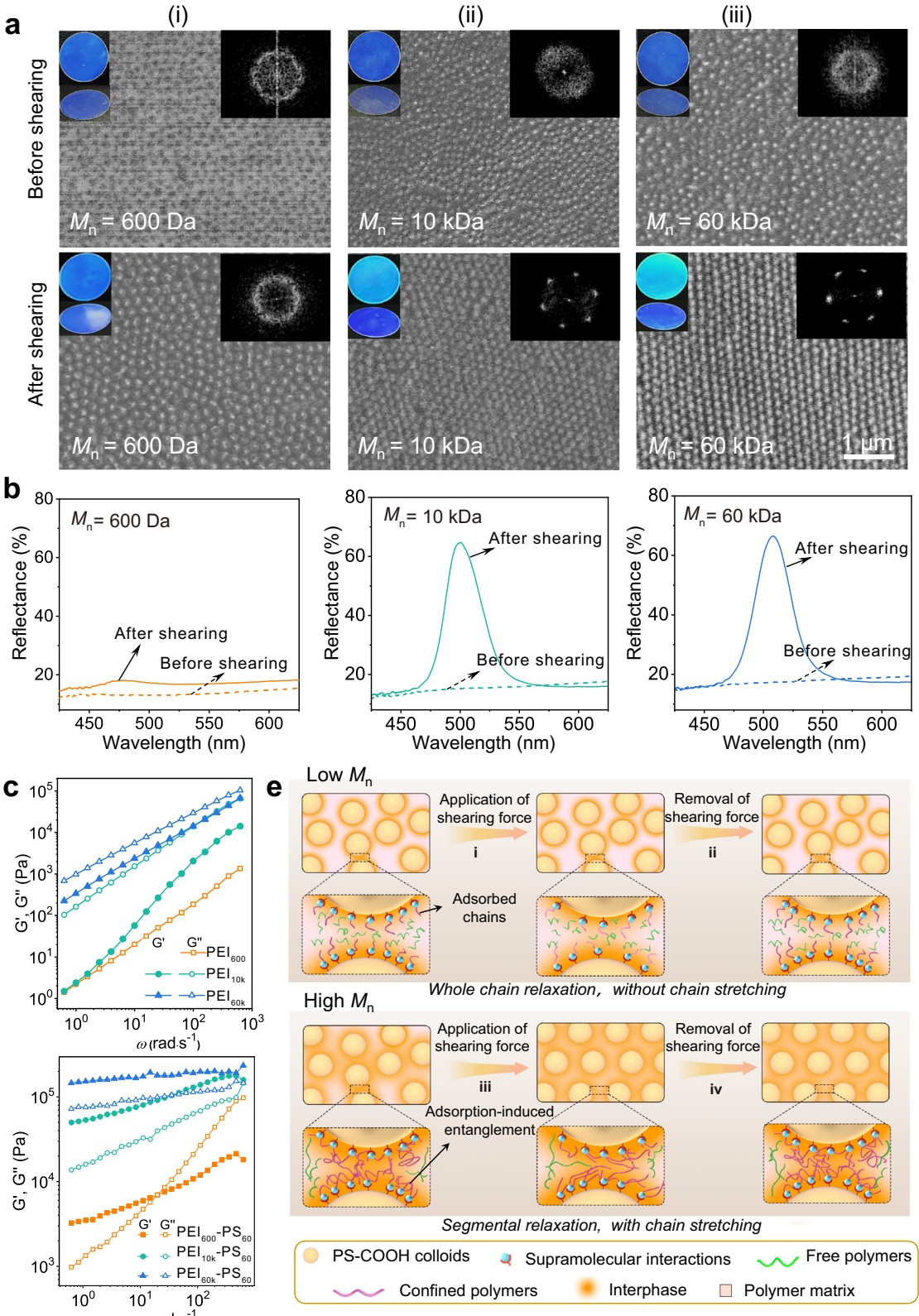

**Fig. 2 | Effect of the $M_n$ of PEI on the shear-induced colloidal ordering effect.**
**a** SEM images of the PEI$_{600}$-PS$_{60}$ (i), PEI$_{10k}$-PS$_{60}$ (ii), and PEI$_{60k}$-PS$_{60}$ (iii) films before (top panels) and after (bottom panels) shearing treatments. Insets in (**a**): corresponding photographs (left) at viewing angles of 0° and 60° and 2D FFT images (right) (sample diameter: 2.8 cm). Here, 0° indicates the direction normal to the film surface. The scale bar in the last image applies to the others. **b** Reflection spectra of the PEI$_{600}$-PS$_{60}$ (left panel), PEI$_{10k}$-PS$_{60}$ (middle panel), and PEI$_{60k}$-PS$_{60}$ (right panel) films before and after shearing treatments. **c**, **d** Dynamic shear moduli (i.e., storage modulus (G′) and loss modulus (G″)) of PEI$_{60}$, PEI$_{10k}$, and PEI$_{60k}$ and their corresponding composites of PEI$_{600}$-PS$_{60}$, PEI$_{10k}$-PS$_{60}$, and PEI$_{60k}$-PS$_{60}$ composites as a function of the angular frequency ($\omega$). The measurements were performed with a strain amplitude of 0.1%. **e** Proposed mechanism of shear-induced colloidal arrangement in the supramolecular composite associated with $M_n$.

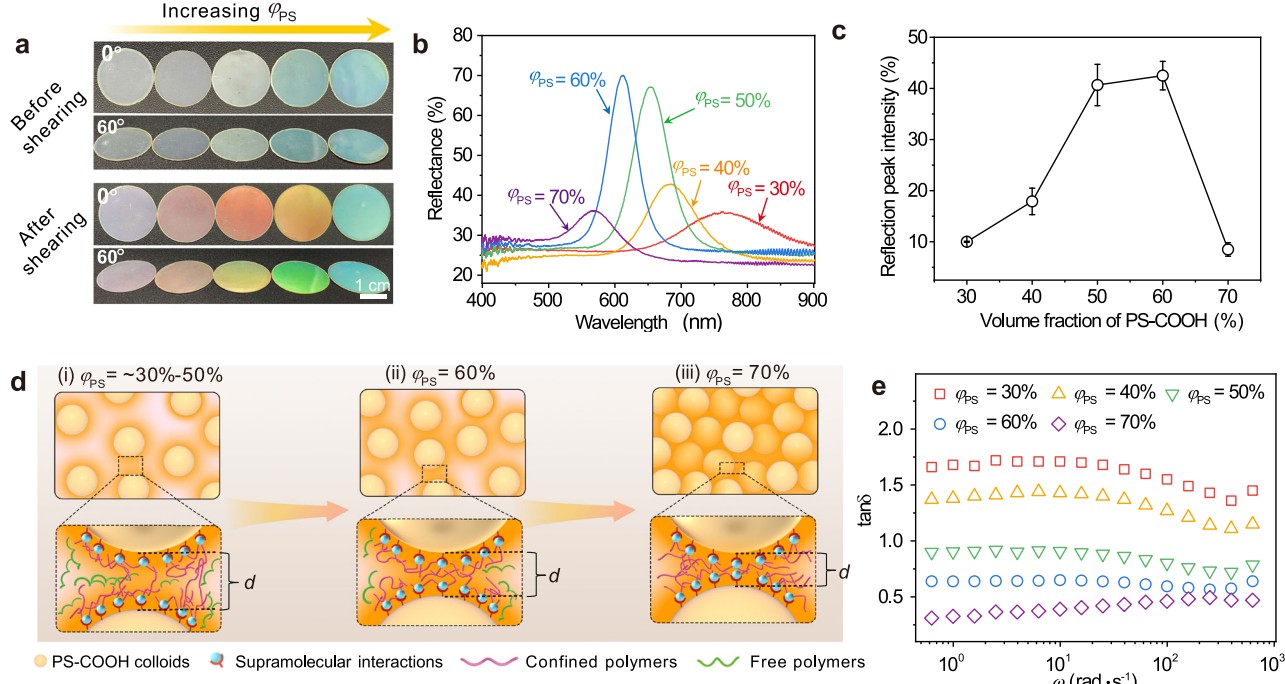

**Fig. 3 | Effect of $\varphi_{PS}$ on the shear-induced colloidal ordering effect.**
**a** Photographs of PEI$_{60k}$-PS$_{30}$, PEI$_{60k}$-PS$_{40}$, PEI$_{60k}$-PS$_{50}$, PEI$_{60k}$-PS$_{60}$, and PEI$_{60k}$-PS$_{70}$ films before and after sufficient shearing treatments at viewing angles of 0° and 60° (sample diameter: 2.2 cm). **b, c** Reflection spectra and the corresponding reflection peak intensity of PEI$_{60k}$-PS$_{30}$, PEI$_{60k}$-PS$_{40}$, PEI$_{60k}$-PS$_{50}$, PEI$_{60k}$-PS$_{60}$, and PEI$_{60k}$-PS$_{70}$ films after sufficient shearing treatments. The reflection peak intensity here is the difference between the reflectivities at the peak and valley in the reflection spectrum. The diameter of the PS-COOH colloids is 203 nm. Error bars represent mean ± standard deviations. $n = 3$ independent experiments. **d** Schematic illustration of the average separation distance ($d$) between two adjacent colloids and the confined state of polymer chains in the supramolecular composite with different $\varphi_{PS}$ values. **e** Loss factor (tanδ) of supramolecular composites with different $\varphi_{PS}$.

$\varphi_{PS}$ values of 30%, 40%, 50%, 60%, and 70%. The laminated composite films (without shearing treatment) with $\varphi_{PS}$ of 30%, 40%, and 50% showed negligible colors (Fig. 3a). In comparison, the films with $\varphi_{PS}$ of 60% and 70% exhibited a faint and angle-independent green color attributed to Mie scattering caused by the high $\varphi_{PS}$[39], which was further supported by their corresponding reflection spectra (Supplementary Fig. 14). After sufficient shearing treatments, the composite films with $\varphi_{PS}$ from 30% to 60% displayed angle-dependent structural colors, with the films of higher $\varphi_{PS}$ exhibiting more intense colors (Fig. 3a). Moreover, the reflection peak wavelength blueshifted from 766 to 683, 653, 611, and 569 nm for the films with $\varphi_{PS}$ from 30% to 70% (Fig. 3b), which was attributed to the reduction in the lattice spacing of the ordered stacking of colloids with increasing $\varphi_{PS}$[30]. The observed colors were consistent with their reflection wavelengths. We compared the reflection peak intensities of the sufficiently sheared composite films to evaluate the ordering of colloidal arrangements within the polymeric matrix. As $\varphi_{PS}$ increased from 30% to 60%, the intensity of the reflection peak increased from 10% to 43% (Fig. 3c), indicating an increased ordering of colloidal arrangements within the film[33]. This result can be attributed to two aspects. On the one hand, the colloidal volume fraction determined the average separation distance ($d$) between two adjacent colloids in the composite[33,58] (Fig. 3d), and $d$ can be expressed using Eq. 1:

$$d = \left(\frac{4\sqrt{2}\pi}{3}\right)^{\frac{1}{3}} r \times \frac{1}{\varphi_{PS}^{\frac{1}{3}}} - 2r \qquad (1)$$

where $r$ is the radius of the PS-COOH colloid. Therefore, the average separation distances between colloids in the supramolecular composites with $\varphi_{PS}$ values of 30%, 40%, 50%, and 60% were calculated to be 71, 46, 28, and 15 nm, respectively. A low average separation distance

between colloids might facilitate the transfer of colloidal momentum through colloidal collisions during shearing, favoring the ordered arrangement of colloids. On the other hand, the PS-COOH colloids acted as crosslinkers and fillers. Therefore, the volume fraction of PS-COOH colloids determined the confined state of the polymer chains[57,58] (Fig. 3d), affecting the slippage (i.e., viscosity) and chain stretchability (i.e., elasticity) during shearing. These factors, in turn, may influence energy transfer between the polymeric matrix and colloids within the composite. Hence, the loss factor, the ratio of the viscous modulus to the elastic modulus determined through rheological measurements, is considered suitable for estimating the energy transfer mechanism of the composite upon shearing[54]. For the composites with $\varphi_{PS}$ of ~30%–50% (i.e., PEI$_{60k}$-PS$_{30}$, PEI$_{60k}$-PS$_{40}$, and PEI$_{60k}$-PS$_{50}$), FTIR spectra revealed that there were unabsorbed free PEI chains interspersed in the polymeric matrix (Fig. 3d(i), Supplementary Fig. 15, and Supplementary Note 8), leading to lower mechanical strength (79, 31, and 284 kPa, confirmed by the stress–strain tests, Supplementary Fig. 16). Moreover, the loss factors of these composites with $\varphi_{PS}$ of ~30%–50% were all above or close to 1 (Fig. 3e), indicating that the viscous flow behavior of the composites dominates under shearing[54]. When $\varphi_{PS}$ reached 60%, the FTIR spectrum revealed that almost all the chains were trapped by supramolecular adsorption and adsorption-induced topological confinement (Fig. 3d(ii)). Consequently, the composite exhibited moderate mechanical strength (~1 MPa, confirmed by the stress–strain test, Supplementary Fig. 16) and good viscoelasticity[54] (loss factor of ~ 0.63, below 1, Fig. 3e). Based on these results, we can infer that as the $\varphi_{PS}$ in the supramolecular composites increased, the energy required to stretch the polymer chains to drag the colloidal migration during shearing increased. As a consequence, the shear-induced ordering effects were improved with increasing $\varphi_{PS}$.

When $\varphi_{PS}$ reached 70%, a noticeable decrease in the reflection peak intensity was observed (Fig. 3b, c), suggesting a significant decrease in

the ordering of the colloidal arrangement in the composite film[33]. This result might be attributed to the high content of PS-COOH colloids, which increased the confinement of the polymer chains brought about by sufficient crosslinking sites on the colloid surface and the contribution from the rigid colloids (Fig. 3d(iii)), increasing the yield stress of the composite[57]. As shown in the stress–strain curve, the $PEI_{60k}$-$PS_{70}$ composite underwent brittle fracture at a stress of 1.9 MPa (Supplementary Fig. 16). This stress value was higher than the applied shear stress (~0.71 MPa, calculated in the Supplementary Note 6), indicating that under these shearing conditions, the polymer chains were difficult to slide, thereby hardly altering the colloidal arrangement[40]. Therefore, in supramolecular composites, by adjusting the polymer molecular weight and colloidal volume fraction, the energy dissipation of the matrix and the energy transfer between the matrix and colloids under shearing can be modified to obtain SCCFs with optimal colloidal arrangements and optical properties. Additionally, these observations demonstrate the importance of well-designed composite formulations for producing SCCFs with brilliant structural colors. Given these results, we chose PEI with an $M_n$ of 60 kDa and colloidal $\varphi_{PS}$ of 60% to prepare the supramolecular composite for further investigations.

## Color quality and colloidal arrangement of SCCFs

The color quality and colloidal arrangement of the optimal SCCFs, i.e., sufficiently sheared $PEI_{60k}$-$PS_{60}$ films, are investigated. To quantitatively evaluate the color quality, we measured the intensities and the normalized FWHMs of the peaks in the reflection spectra of the sufficiently sheared composite films, corresponding to the brightness and saturation of the visually perceived colors[33]. For example, an SCCF constructed from colloids with a diameter of 192 nm displayed a visually appealing appearance with bright colors and good flexibility (Fig. 4a). Correspondingly, the reflection peak intensity at a 0° viewing angle reached 60%. The normalized FWHM was as low as 0.071 (Fig. 4b). Furthermore, we compared the color quality, i.e., the reflection peak intensity and the normalized FWHM of existing SCCFs constructed by different methods (Fig. 4c). To provide a more accurate and comprehensive comparison of the color quality of SCCFs, the refractive indices of the colloids and polymers in the SCCFs were provided (Supplementary Table 1). We find that the color quality of the current SCCFs obtained by shearing supramolecular composites was comparable to that of SCCFs obtained by shearing core-interlayer-shell colloids and solvation force-induced colloidal assembly methods (Fig. 4c). Significantly, the present method has a significant advantage in producing SCCFs with high-quality structural colors over capillary-force and magnetic-force-induced colloidal assembly methods (Fig. 4c). This implies that the shearing supramolecular composite method is effective in preparing SCCFs with high color brightness and saturation. Notably, such a structural color stems from the Bragg diffraction of visible light and can be precisely tailored by employing colloids with various diameters[59]. SCCFs composed of colloids with various diameters produced bright blue, cyan, green, yellow, red, and magenta colors after shearing treatments (Fig. 4d). All the films presented an angle-dependent peak shift of ~50 nm when observed at angles from 0° to 60° (Supplementary Fig. 17). These brilliant iridescent colors were attributed to the well-arranged colloids within the SCCFs induced by shearing, despite the low refractive index contrast of 0.061 between the PS-COOH colloids (1.590) and the PEI polymers (1.529).

To investigate the colloidal arrangement within the composite film, we directly observed the top and bottom surfaces and the cross-section of the sufficiently sheared composite film using SEM imaging. The colloids on both the top and bottom surfaces of the composite film exhibited ordered hexagonal packing arrangements. These hexagonal packing arrangements adopted different orientations similar to multiple domains along with numerous dislocations (white hexagons outlined in Fig. 4e). Moreover, the ordering of the colloidal arrangements at the bottom of the composite film, i.e., near the rod surface, decreased in

the thickness direction and broke into fragments of hexagonal stacking arrangements at ~62 μm (Fig. 4f(iii)). This result could be attributed to the different shear strains experienced by the composite film along with its thickness during shearing[60]. Nevertheless, the decreased ordering in the top surface of the SCCF did not significantly impact its color quality. This is because both reflectance spectroscopy and visual inspection have a certain detection depth for evaluating the color quality of SCCF. The number of stacking layers ($d'$) in the surface planes of the sample that can be detected by reflection spectroscopy depends on the refractive index contrast ($\triangle n$) and effective refractive index ($\bar{n}$) of the composites[37], which can be expressed as $d' = \bar{n}/\triangle n$. Therefore, for SCCF with a refractive index contrast of 0.061, the reflectance spectrometer can detect up to 26 layers of colloidal arrangement, corresponding to a detection depth of SCCF less than 6 μm. The SEM image in Fig. 4f shows that SCCF exhibited a highly long-range ordered arrangement with a depth of up to 32 μm. This indicates that although there are some defects in the colloidal arrangement on the top surface of SCCF, the structural color with high brightness and saturation can still be observed by the naked eye and analyzed by reflectance spectroscopy. These results demonstrate that with the aim of achieving high-quality structural colors, SCCFs with well-arranged colloids were successfully prepared. Additionally, the enhancement of characteristic peaks in the one-dimensional small-angle X-ray scattering curve of the sheared composite film also suggested an overall improvement in the ordering of colloidal arrangements after shearing treatment[38] (Supplementary Fig. 18).

To better understand the shear-induced ordering process, we compared the colloidal arrangements of insufficiently sheared films: a film that was insufficiently sheared for only a few ordered layers at the bottom surface (Supplementary Fig. 19). In contrast, a sufficiently sheared film exhibited ordered layers throughout the entire film thickness (Fig. 4e). These results suggest that the ordered arrangement of the colloids in the SCCF produced by shearing developed layer-by-layer from the bottom surface toward the top. Overall, SCCFs obtained by shearing supramolecular composites exhibit finely colloidal arrangements and brilliant colors, thus positioning them as highly promising alternatives to traditional dyes.

## Processing advantages of the supramolecular composite

In particular, we are intrigued by the remarkable processing performance of the supramolecular composite. The intricate dance between solvents and temperature variations plays a pivotal role in the dissociation of supramolecular interactions, paving the way for enhanced processability and ease of manufacturing[50,61]. Exploiting the vast diversity of these supramolecular interactions unlocks a treasure trove of possibilities for expanding the horizons of manufacturing SCCFs[47]. To meticulously examine the impact of the water content on the processability, a series of $PEI_{60k}$-$PS_{60}$ composites with different water contents were prepared. We found that for the $PEI_{60k}$-$PS_{60}$ composites with water content of 10 wt.%, 15 wt.%, and 20 wt.%, the reflection peak intensity after sufficient shearing can reach 60%, while for $PEI_{60k}$-$PS_{60}$ composites with water content as high as 25 wt.% and 30 wt.%, the reflection peak intensity significantly decreased to 37% and 4%, respectively (Supplementary Fig. 20). This may be due to the gradual deterioration of the elastic properties at high water content, which hinders the transfer of colloidal momentum during shearing. Based on these results, the processing performance of $PEI_{60k}$-$PS_{60}$ composites with water content of 10 wt.%, 15 wt.%, and 20 wt.% was further studied in detail. To ensure that the stable ordered colloidal arrangements generated by shearing treatment in these composites could be maintained, we evaluated the thermal motion ability of colloids under static conditions (i.e., without shearing forces) by calculating the diffusion coefficients of colloids in composites with different water contents (Supplementary Note 9). It was found that the diffusion coefficients of colloids in composites were $4.06 \times 10^{-21}$, $7.03 \times 10^{-21}$, and $2.67 \times 10^{-20}$

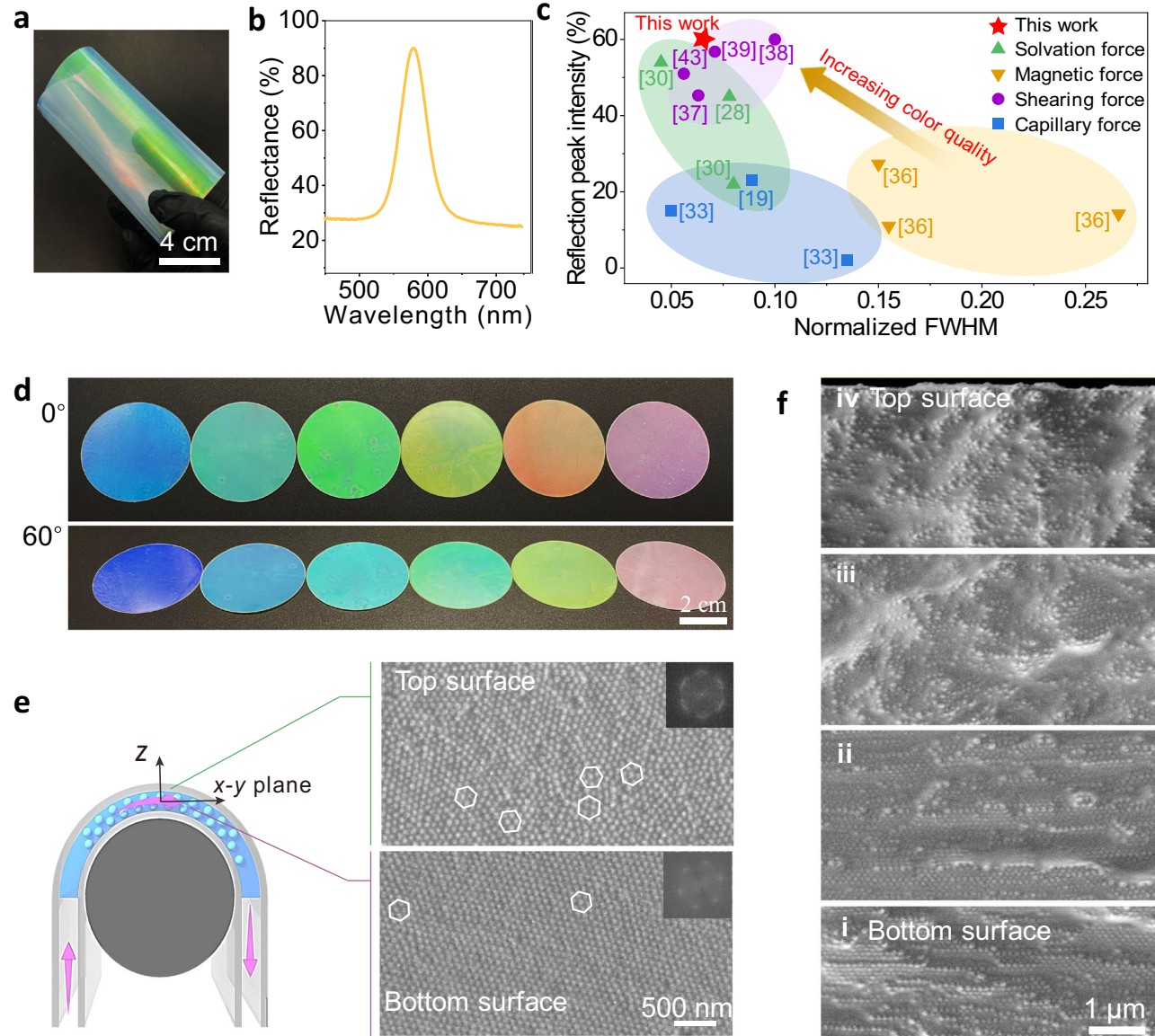

**Fig. 4 | Color quality and colloidal arrangement of optimal SCCFs obtained by shearing supramolecular composites. a** Photograph of an SCCF (i.e., sufficiently sheared $PEI_{60k}$-$PS_{60}$ film) (sample size: 21 × 13 cm²). **b** Reflection spectrum of the SCCF. Note that the photograph and reflection spectrum were recorded on the top surface of the SCCF. **c** Comparison of reflection peak intensity and normalized full width at half maximum (FWHM) of SCCFs in this study with those prepared by evaporation-induced colloidal assembly (including capillary and solvation force-induced colloidal assembly methods) and external force-induced colloidal arrangement (including magnetic force-induced colloidal arrangement and shearing core-interlayer-shell structured colloid). The normalized FWHM is the ratio of the FWHM to the reflection wavelength. The shadow represents the color quality range of SCCF obtained by different strategies, with the purple representing the color quality of SCCF obtained by shear-induced colloidal arrangement, and the yellow, blue, and green representing the color quality of SCCF obtained by magnetic force, capillary force, and solvation force induced colloidal assembly, respectively. The numbers represent the reference numbers. **d** Photographs of SCCFs with different colors constructed from colloids with various diameters at viewing angles of 0° and 60°. The diameters of the colloids used for preparing the different SCCFs from left to right are 177, 182, 188, 192, 203, and 225 nm (sample diameter: 4.4 cm). **e** Illustration showing the top and bottom surfaces of the composite film during the bending-induced shearing treatment and SEM images of the corresponding top and bottom surfaces of the SCCF after sufficient shearing treatment, where the white hexagons outline hexagonal packing colloidal arrangements with different orientations. Insets: corresponding 2D FFT images. The bottom surface faced the rod. The scale bar in the bottom image applies to the top image. **f** Cross-sectional SEM images taken along the thickness direction from the composite film bottom to the top at distances of ~ 1 μm (i), ~ 32 μm (ii), ~ 62 μm (iii), and ~ 106 μm (iv). The scale bar in the last image applies to the others.

m²·s⁻¹, respectively. These low diffusion coefficients suggest that the ordered arrangements generated by shearing force can be retained in these high-viscosity composites even after removing shearing forces[33]. Then, we evaluated their processing performance by measuring the intensity of the reflection peak of the composite film after different numbers of shearing passes. The $PEI_{60k}$-$PS_{60}$ composites with 10 wt.%, 15 wt.%, and 20 wt.% water content exhibited uniform green appearances with increased brightness after the same number of shearing passes, with reflection peak intensities of 18%, 31%, and 60% (Fig. 5a, b). Moreover, we found that under the same shearing conditions, the composite with 20 wt.% water required 20 shearing passes to yield an SCCF with a reflection peak intensity of ~60%. In contrast, the composites with 10 wt.% and 15 wt.% water required 50 and 40 passes, respectively (Fig. 5c). These results suggested that the introduction of water made composite processing milder and more productive and facilitated the industrial production of SCCFs.

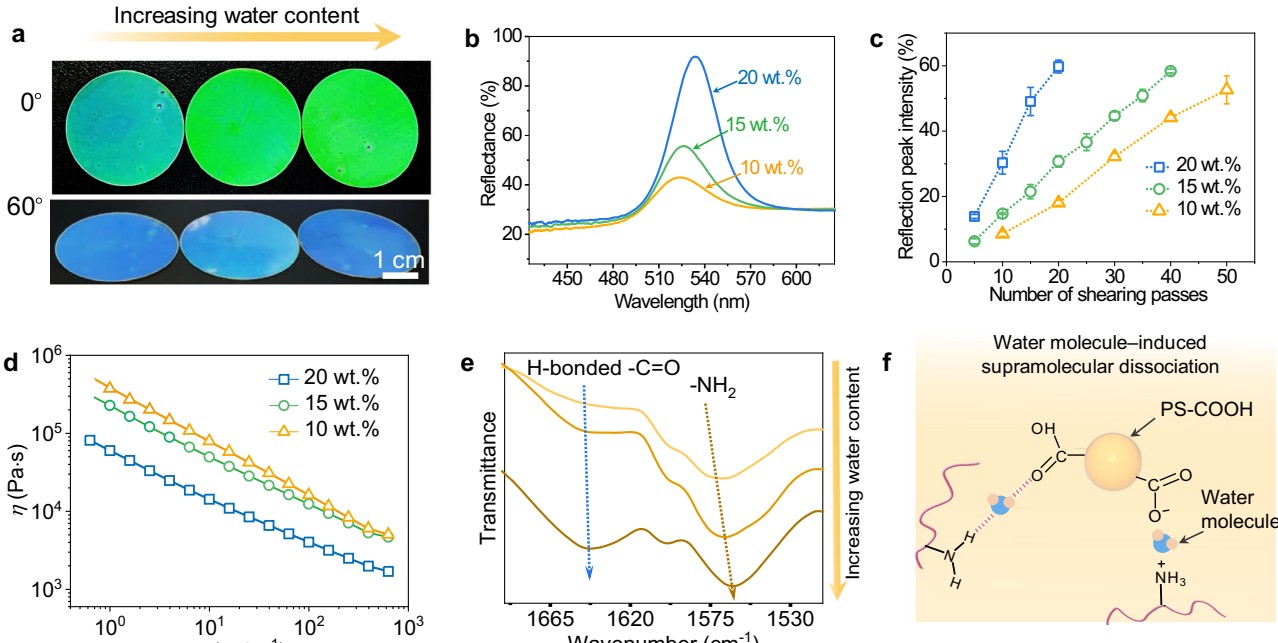

**Fig. 5 | Effect of water on the processing performance of the supramolecular composite. a**, **b** Photographs and reflection spectra of $PEI_{60k}$-$PS_{60}$ films with water contents of 10 wt.%, 15 wt.%, and 20 wt.% after 20 shearing passes (sample diameter: 3.4 cm). The diameter of the rod used here is 6 mm. **c** Reflection peak intensity of $PEI_{60k}$-$PS_{60}$ films with different water contents as a function of the number of shearing passes. Error bars represent mean ± standard deviations. $n = 3$

independent experiments. **d** Complex viscosity ($\eta$) of $PEI_{60k}$-$PS_{60}$ composites with water contents of 10 wt.%, 15 wt.%, and 20 wt.% as a function of frequency. **e** FTIR spectra of $PEI_{60k}$-$PS_{60}$ films with different water contents. **f** Proposed mechanism for water molecules to reduce the viscosity of supramolecular composites due to the dissociation of hydrogen bonding and electrostatic interactions.

To gain an in-depth understanding of the effect of water on processing performance, we used rheology techniques to investigate the mechanical behavior of the composites with different water contents. The results showed that the viscosity and shear modulus of the $PEI_{60k}$-$PS_{60}$ composites significantly decreased with increasing water content (Fig. 5d, Supplementary Fig. 21, and Supplementary Note 10). In addition, FTIR spectra of $PEI_{60k}$-$PS_{60}$ composites with varying water content were measured to investigate the impact of water molecules on supramolecular interactions. As the water content increased, the stretching vibration peaks of -C = O involved in hydrogen bonding and the bending vibration peaks of -$NH_2$ groups, shifted to lower wavenumbers (Fig. 5e). This observation suggests the formation of hydrogen bonding between water molecules and -C = O and -$NH_2$ groups. As a result, the hydrogen bonds between -$NH_2$ groups on the PEI and -C = O groups on the surface of PS-COOH were dissociated[50] (Fig. 5f). The viscosity and the modulus of the composites decreased significantly, which implies a reduction in the viscous resistance to colloidal migration during the shearing process, thereby improving the processing performance of the composite.

Elevated temperature, promoting supramolecular dissociation, can also improve the processing performance of the composites[60]. The quantitative effect of processing temperature on the processing performance of supramolecular composites, including the optical (Supplementary Fig. 22 and Supplementary Note 11) and mechanical properties (Supplementary Fig. 23 and Supplementary Note 12) of composites under different processing temperatures, along with the corresponding validation example (Supplementary Fig. 24 and Supplementary Note 13), were investigated. These results demonstrate that high-modulus supramolecular composites, which are typically challenging to process at room temperature, can be successfully processed into films with high-quality structural colors by slightly raising the temperature.

The processing advantages offered by supramolecular composites and the diversity of supramolecular interactions inspired us to

explore using various commonly-used polymers with wider $T_g$ ranges to construct SCCFs with high-quality structural colors. A series of polymers with a wide range of $T_g$ values were selected (Supplementary Fig. 25 and Supplementary Note 14), including commercially available polyacrylic acid (PAA, $T_g$: 115 °C), polyvinylpyrrolidone (PVP, $T_g$: 95 °C), poly(diallyldimethylammonium chloride) (PDDA, $T_g$: 45 °C), PEG-*b*-PPG-*b*-PEG ($T_g$: −67 °C), as well as synthetic polyurethane (PU, $T_g$: −63 °C) and poly(borosiloxane) (PBSi, $T_g$: −123 °C) (Fig. 6a). These polymers can form supramolecular composites with typical colloids such as PS-COOH or silica ($SiO_2$) colloids. A series of results show that by introducing an appropriate amount of solvent, these composites constructed from different $T_g$ polymers could be processed into films with brilliant colors at room temperature (Supplementary Fig. 12 and Supplementary Figs. 26–28). To further show the advantages of shearing supramolecular composites over previously reported shearing strategies, we compared the processing performance of shear-induced colloidal arrangements in implementing colloids with core-interlayer-shell structures and supramolecular composites from the perspectives of the processing temperature and $T_g$ of the matrix polymer. The mild processing conditions associated with the supramolecular composites, coupled with the broader selection range of matrix polymers exhibiting different $T_g$ values (Fig. 6b), clearly demonstrate the universality of material selection and processing friendliness of the current method in producing SCCFs. Moreover, we found that SCCFs obtained from shearing supramolecular composites possessed stable optical properties at room temperature and can be cyclically reused with the assistance of a suitable solvent, which further enhances the practical applicability of SCCFs (Supplementary Figs. 29 and 30). Additionally, a comprehensive comparison was made between our method and previous approaches for constructing SCCFs, including capillary and solvation force-induced colloidal assembly methods, the magnetic force-induced colloidal arrangement, and shearing core-interlayer-shell structured colloids by comparing

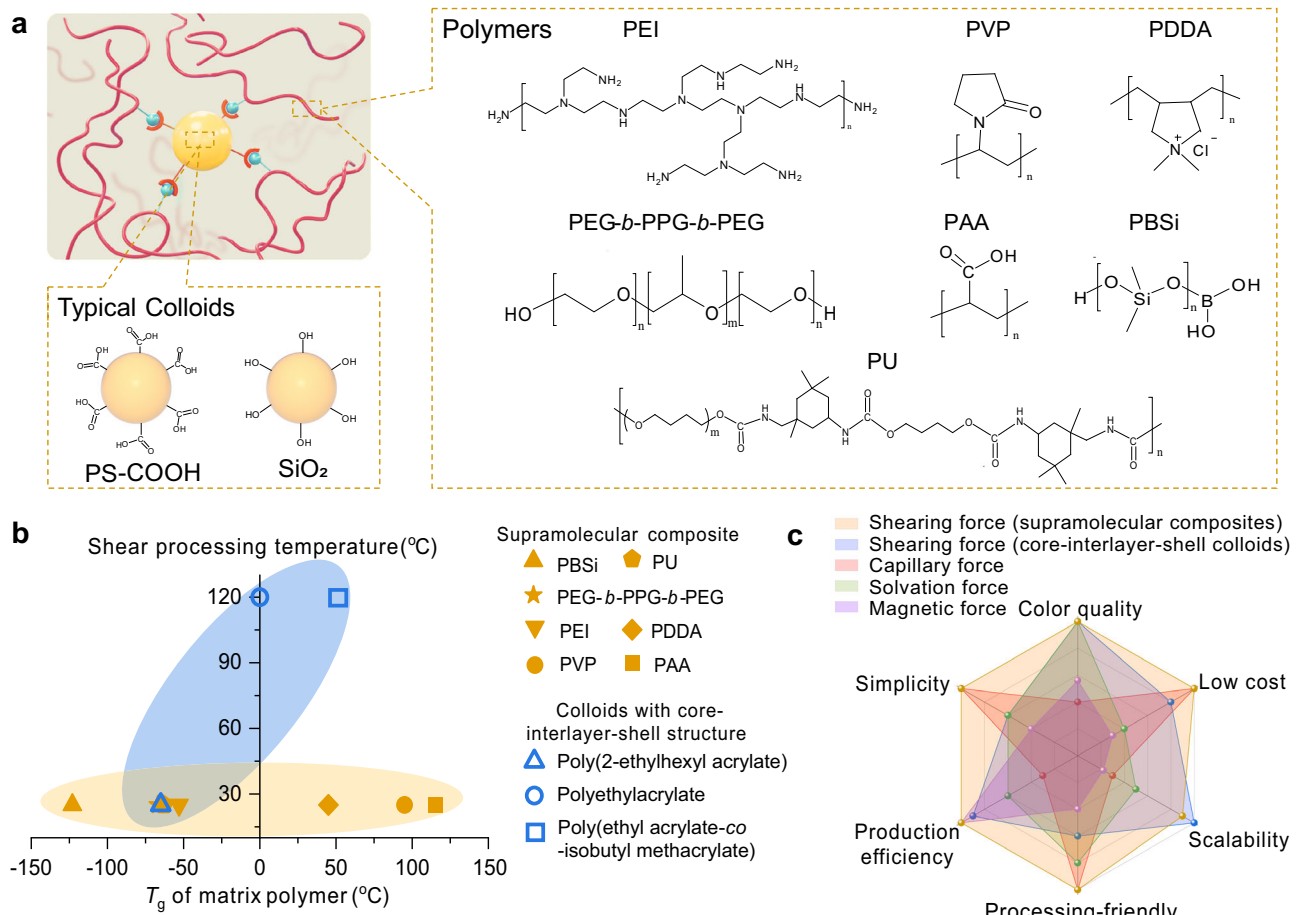

**Fig. 6 | The versatility of shearing supramolecular composite. a** Chemical structures of commonly-used polymers (e.g., PEI, polyvinylpyrrolidone (PVP), poly(diallyldimethylammonium chloride) (PDDA), poly(ethylene glycol)-*block*-poly(propylene glycol)-*block*-poly(ethylene glycol) (PEG-*b*-PPG-*b*-PEG), poly(-borosiloxane) (PBSi), polyacrylic acid (PAA), and polyurethane (PU)) and typical colloids with functional groups (e.g., PS-COOH and SiO₂) for constructing SCCFs. **b** Comparison of the processing temperature and $T_g$ of the matrix polymer in supramolecular composites and colloids with core-interlayer-shell structures for constructing SCCFs. The shading represents the corresponding shear processing temperature intervals for supramolecular composites (yellow region) and colloids with core-interlayer-shell structures (blue region) composed of different $T_g$ matrix polymers. **c** Comparison between evaporation-induced colloidal assembly (including capillary and solvation force-induced colloidal assembly methods) and external force-induced colloidal arrangement (including magnetic force-induced colloidal arrangement and shearing force-induced colloidal arrangement implemented in core-interlayer-shell structured colloid and supramolecular composite methods) for the construction of SCCFs.

the materials used, production efficiency, processing conditions, preparation steps, and production area for each strategy. The shearing technique employed in our supramolecular composites boasts several notable advantages, including low cost, high production efficiency, processing friendliness, simplicity, and scalability, making it an ideal choice for the fabrication of optical materials with high-quality colors, such as colorful coatings, decoration of phone cases, and stylish wristbands (Fig. 6c, Supplementary Table 2, and Supplementary Fig. 31). As proven above, the customized roll-to-roll setup used in this study allowed for continuous shearing processing of the supramolecular composites, enabling the production of large-scale SCCFs with ease. This rapid, mild, and large-scale production of SCCFs opens up great possibilities for commercial applications of these optical materials. Furthermore, this method shows potential for expanding to the precise orientation of one-dimensional or two-dimensional nanofillers, providing significant opportunities for creating a wide range of functional structural composite materials suitable for various applications.

## Discussion

We have developed a mild shearing method to construct meter-scale SCCFs with high-quality structural colors by shearing supramolecular composites composed of commonly-used polymers and colloids. We

found that the polymer molecular weight and colloid volume fraction in the supramolecular composite played crucial roles in shear-induced colloidal ordering and elucidated the intrinsic mechanisms, providing significant guidance for designing SCCFs with desired structural colors. Furthermore, due to the dynamic reversibility of supramolecular interactions, the processing performance can be significantly improved by introducing solvents or raising processing temperature, resulting in the fabrication of more processable SCCFs. Notably, this method exhibits excellent versatility in preparing SCCFs using various commonly available polymers and colloids, thus expanding the range of material choices. The obtained SCCFs exhibit brilliant colors and promise as alternatives to conventional energy-intensive, polluting, and easily degradable dyes. As such, we anticipate this method will provide a mild manufacturing avenue for developing advanced structurally composite materials with broad practical applications.

## Methods

### Materials

Styrene (St, 99.5%), acrylic acid (AA, 98%), and ammonium persulfate (APS, 98%) were purchased from Sinopharm Chemical Reagent Co., Ltd. PEIs with $M_n$ values of 600 Da, 10 kDa, 60 kDa, and 2000 kDa were purchased from different sources of Innochem, Sigma Aldrich, and

Wuhan Lullaby Pharmaceutical Chemical Co., Ltd.. All chemicals were of analytical grade and used as received.

## Synthesis of PS-COOH colloids
PS-COOH colloids were synthesized by an emulsifier-free copolymerization method[33]. A total of 960 mL $H_2O$, 60 mL St, and 16 mL AA were mixed in a 2 L three-necked flask equipped with a reflux condenser. Then, 1.0 g APS was added to the above mixture solution, and the mixture was stirred for 2 h at 90 °C under an $N_2$ atmosphere. The PS-COOH colloids were separated by centrifugation and washed with ethanol three times. PS-COOH colloids with different sizes were synthesized by tuning the amount of styrene and ammonium persulphate.

## Preparation of supramolecular composites
Supramolecular composites were prepared by a facile blending of colloid dispersions and polymer solutions, followed by solvent removal. Taking PEI-PS composites as an example, PS-COOH was first dispersed in water and then mixed with PEI solution to obtain the precursor dispersions. The precursor dispersion was continuously stirred at 80 °C, allowing water evaporation to obtain a supramolecular composite PEI-PS. Various composites were constructed by varying the molecular weight of PEIs and the volume fraction of PS-COOH and named $PEI_x-PS_y$, where $x$ is the $M_n$ value of PEI and $y$ is the volume fraction of PS-COOH.

## Characterizations
Scanning electron microscopy (SEM) images were taken by a HITACHI SU8010 SEM at an acceleration voltage of 1 kV. Reflection spectra were measured with a USB4000 fiber optical spectrometer (Ocean Optics). Polished silicon wafers were selected as the standard reflector for reflectivity. Digital photographs were taken on a Canon PowerShot SX6 camera. Nonlinear oscillatory shear rheology measurements were performed on a HAAKE MARS60 rheometer (Germany) with parallel plate geometry (20 mm diameter rotating top plate) and a strain amplitude of 0.1% over a frequency range of 0.1–100 Hz. Fourier transform infrared (FTIR) spectra were measured on an Equinox 55 spectrophotometer (Bruker) equipped with an attenuated total refraction attachment with a horizontal ZnSe crystal.

## Data availability
All data needed to evaluate the conclusions in the paper are present in the paper and the Supplementary Information. Data is available from the authors on request. Source data are provided with this paper.

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

## Acknowledgements

This research was supported by the National Natural Science Foundation of China (52373075 and 52022032) (L.Z.) and the China Postdoctoral Science Foundation (2022M711246) (M.L.). We also thank the HUST Analytical and Testing Center for their help with the facilities.

## Author contributions

L.Z. conceived and supervised the project. M.L. performed the experimental studies with help from B.P., Q.L., X.C., Z.H., X.Z., B.X., L.Z. and J.Z. All authors revised and edited the paper.

## Competing interests

The authors declare no competing interests.
