## [Peer Review File · Nature Communications]

Scalable production of structurally colored composite films by shearing supramolecular composites of polymers and colloidsREVIEWER COMMENTS

Reviewer #1 (Remarks to the Author):

The authors present a general and mild roll-to-roll shearing approach to produce large-scale structural colored composite films (SCCFs) by shearing supramolecular composites composed of carboxylated polystyrene (PS-COOH) colloids and polyethyleneimine (PEI) with supramolecular interactions. The resultant flexible SCCFs in this study exhibited high color brightness and saturation, with a reflection peak intensity of 60% and a narrow full width at half maximum (FWHM) of 43 nm. The fabrication method is scalable, and the authors have provided data to show the universality of the approach in commonly-available polymer materials. However, the results presented in the manuscript have some major issues, and also lacks novelty in the methods. Please find my comments below.

1. The roll-to-roll shearing approach to produce large-scale structural color films has also been reported at Nat. Commun. 7, 11661 (2016); Materials 10, 688 (2017); J Colloid Interf. Sci. 584, 145-153 (2021); Adv. Sci. 9, 2202061 (2022).
2. Refractive index contrast is a key parameter to determine the reflectivity and brightness of photonic crystals. However, how does the quite low contrast between PEI polymer and PS colloid guarantee the high color brightness and saturation in this work.
3. The author claimed in abstract “producing SCCFs with well-arranged colloids is bottlenecked.” However, the bottleneck is not seemed to break in this work. Obvious defeats could be noticed in SEM images in colloidal arrangement.
4. The authors provide a complete survey and comparison to support that their proposed approach has the advantages of low cost, high production efficiency, processing friendliness, simplicity, and scalability, do the authors have any data supporting?
5. The application of the flexible SCCFs could be considered to explore.
6. Please add scale in Fig. 4a, Fig. 4d, and Fig. 1d for reference.
7. The authors investigate the role of polymer molecular weight on shear-induced colloidal ordering, why are higher molecular weights of PEI polymer not chosen (more than 60 k).

Reviewer #2 (Remarks to the Author):

In the manuscript by Li et al., the authors reported on a strategy to produce large-scale structurally colored composite films by shearing supramolecular composites composed of polymers and colloids with supramolecular interactions. The authors effectively exploit the supramolecular connections between polymers and colloids, enabling the ordered arrangement of colloids through efficient momentum transfer. Furthermore, the dynamic nature of supramolecular interactions is harnessed to enhance the processability of the composite. The authors also provided a comprehensive elucidation of the mechanism behind shear-induced colloidal ordering and demonstrated the remarkable generality and versatility of this approach. This work provided an innovative approach to producing large-scale

structurally colored composite films with attractive properties, which are challenging. Overall, the manuscript is well written, and I believe it will appeal to the broad readership of Nature Communications. I recommend publication after the following questions have been clarified.

1. The authors elucidated that the shear-induced colloidal ordering in supramolecular composites is related to the molecular weight of PEI. Therefore, understanding the rheological properties of pure PEI is critical when constructing structurally colored composite films using shearing methods. It is recommended to add this information to the main text and discuss it further.

2. Can these samples be recycled as the composites are obtained based on supramolecular interactions? In addition, how is the stability of the structure or optical properties of structurally colored composite films?

3. In Figure 4c, the authors compellingly demonstrate the superiority of the shearing supramolecular composite strategy by comparing the optical properties of structurally colored composite films obtained through different approaches, using reflection peak intensity and normalized FWHM as metrics. However, it is crucial to acknowledge that the optical quality of structurally colored composite films is intricately linked to the material's refractive index. Therefore, it is strongly recommended to include the refractive index of the polymer matrix and colloid, even if provided in the Supplementary Information (SI) section, to ensure a comprehensive evaluation of the optical characteristics.

4. Characterization of Colloid Size and Distribution: The manuscript lacks information on the size and distribution of colloids.

5. Impact of Water Content (More than 20 wt%): The manuscript should address the potential impact of high water content (more than 20 wt%) on processability. The optimal upper limitation of water content is not clear.

Point-by-point response to the reviewers' comments for manuscript NCOMMS-23-47461

Reviewer #1 (Remarks to the Author):

The authors present a general and mild roll-to-roll shearing approach to produce large-scale structural colored composite films (SCCFs) by shearing supramolecular composites composed of carboxylated polystyrene (PS-COOH) colloids and polyethyleneimine (PEI) with supramolecular interactions. The resultant flexible SCCFs in this study exhibited high color brightness and saturation, with a reflection peak intensity of 60% and a narrow full width at half maximum (FWHM) of 43 nm. The fabrication method is scalable, and the authors have provided data to show the universality of the approach in commonly-available polymer materials. However, the results presented in the manuscript have some major issues, and also lacks novelty in the methods. Please find my comments below.

Response: Thanks for the Reviewer's insightful and affirmative comments on the scalability and universality of our approach, which has encouraged us to improve our work.

1. The roll-to-roll shearing approach to produce large-scale structural color films has also been reported at Nat. Commun. 7, 11661 (2016); Materials 10, 688 (2017); J Colloid Interf. Sci. 584, 145-153 (2021); Adv. Sci. 9, 2202061 (2022).

Response: We thank the Reviewer for carefully assessing our manuscript. We fully agree with the Reviewer that 'The roll-to-roll shearing approach' has been employed to construct structurally colored films. However, in the works mentioned by the Reviewer (or other works mentioned in our original manuscript), although the roll-to-roll shearing methods were used, building blocks for these structural color films were unfortunately limited to some specific core-shell structured colloids or cellulose nanocrystals and these structural color films were generally obtained at high temperatures with restricted material systems in these existing studies. In sharp contrast, we proposed new material systems of supramolecular slurries suitable to the roll-to-roll shear processing to obtain structural color composite films (SCCFs) with a broad range of colloids and polymers based on rationally optimized supramolecular interactions. Therefore, we would like to emphasize that the novelty of the present

manuscript, compared to previous works, lies in using supramolecular strategies to address the challenges of large-area and mild preparation of SCCFs. The diversity and tunability of supramolecular interactions allow SCCFs with high-quality structural colors to be obtained with commonly available polymers and colloids. Additionally, the dynamic reversibility of supramolecular interactions enables the processing of composite materials formed by polymers with high modulus or glass transition temperature at room temperature. In addition, we have comprehensively described the mechanism of shear-induced colloidal ordering in supramolecular composites, offering a thought-provoking approach for the mild and large-scale preparation of ordered composites via a supramolecular strategy. We believe that these aspects constitute a significant scientific and technological advance over prior work.

Thanks to the Reviewer's comments, to further highlight our innovations compared with these existing studies, we have added some notes in the revised manuscript, as read: "This study overcomes the traditional material limitations of manufacturing SCCFs by shearing method and opens an avenue for mildly producing ordered composites with commonly-available materials via supramolecular strategies." in the Abstract on page 1 and "To further show the advantages of shearing supramolecular composites over previously reported shearing strategies, we compared the processing performance of shear-induced colloidal arrangements in implementing colloids with core-interlayer-shell structures and supramolecular composites from the perspectives of the processing temperature and T_g of the matrix polymer. The mild processing conditions associated with the supramolecular composites, coupled with the broader selection range of matrix polymers exhibiting different T_g values (Fig. 6b), clearly demonstrate the universality of material selection and processing friendliness of the current method in producing SCCFs." on page 23.

2. Refractive index contrast is a key parameter to determine the reflectivity and brightness of photonic crystals. However, how does the quite low contrast between PEI polymer and PS colloid guarantee the high color brightness and saturation in this work.

Response: Thanks for the Reviewer's insightful questions. Refractive index contrast is indeed a key parameter that might affect the reflectivity (i.e., brightness) of photonic materials. However, in practice, the ordered arrangement of colloids is a prerequisite for photonic materials to obtain high reflectivity or bright structural colors. Our method of shear-induced colloidal ordering in supramolecular

composites offers a mild and large-scale preparation of SCCFs with well-ordered arrangements, allowing for high reflectivity. Similar results of well-ordered arrangements of colloids in polymeric matrix with a relatively low refractive index contrast have also been observed. For example, Kim et al. constructed SCCFs with reflectivity up to 60% using poly(ethylene glycol) phenyl ether acrylate polymers and silica colloids as the building blocks with a refractive index contrast as low as 0.052 by improving the ordering of the colloidal arrangement [ACS Nano 11, 11350-11357 (2017)]. Wu et al. constructed SCCFs with high color saturation with a narrow half-peak width of 20.7 nm using poly(dimethylsiloxane) polymers and poly(methyl methacrylate) colloids as building blocks with a refractive index contrast as low as 0.08 [Nanoscale 11, 13377-13384 (2019)]. These facts suggest that building blocks with a low refractive index contrast can achieve high color brightness and saturation in an SCCF with well-ordered arrangements. In our study, we used PEI and PS colloids as the building blocks, with their refractive indices being 1.529 and 1.590, respectively, resulting in a refractive index contrast of 0.061, greater than that in Kim's study (0.052). Therefore, SCCF with a low refractive index difference can obtain structural colors with high brightness and saturation by shear-induced colloidal ordering.

We appreciate the Reviewer's question and have added some notes to explain the high color brightness and saturation of the SCCF in the revised manuscript, as read: “These brilliant iridescent colors were attributed to the well-arranged colloids within the SCCFs induced by shearing, despite the low refractive index contrast of 0.061 between the PS-COOH colloids (1.590) and the PEI polymers (1.529).” on page 17.

3. The author claimed in abstract “producing SCCFs with well-arranged colloids is bottlenecked.” However, the bottleneck is not seemed to break in this work. Obvious defects could be noticed in SEM images in colloidal arrangement.

Response: Thanks for the comments. As pointed out by the Reviewer, there are indeed some defects that can be observed on the top surface of the SCCFs obtained through shearing supramolecular composites. This phenomenon can be attributed to the large thickness of the composite film, which leads to a significant shear strain on the top surface of the SCCF. As a result, the colloidal arrangement on the top surface of the SCCF may undergo multiple orientations, resulting in some defects. This observation is consistent with those in reported literature on shear-induced colloidal ordering [Nat.

Commun. 7, 11661 (2016); Materials 10, 688 (2017)]. However, the defects on the top surface of SCCF did not significantly impact its structural color quality. This is because both reflectance spectroscopy and visual inspection have a certain detection depth for evaluating the optical properties of SCCF. The number of stacking layers (d') in the surface planes of the sample that can be detected by reflection spectroscopy depends on the refractive index contrast (Δn) and effective refractive index (\bar{n}) of the composites, which can be expressed as $d' = \bar{n}/\Delta n$. Therefore, for SCCF with a refractive index contrast of 0.061, the reflectance spectrometer can detect up to 26 layers of colloidal arrangement, corresponding to a detection depth of SCCF less than 6 μm . The SEM image in Fig. 4f in the revised manuscript shows that SCCF exhibited a highly long-range ordered structure with a depth of up to 32 μm . This indicates that although there are some defects in the colloidal arrangement on the top surface of SCCF, the structural color with high brightness and saturation can still be observed by the naked eye and analyzed by reflectance spectroscopy.

Furthermore, by comparing the color quality of SCCFs obtained through shearing supramolecular composites with other SCCFs obtained through different methods in this field, we find that the SCCF prepared by shearing supramolecular composites exhibited outstanding color quality (Fig. 4c for comparison results in the revised manuscript), indicating a highly ordered colloidal arrangement. Based on these results, we believe that with the aim of achieving high-quality structural colors in SCCFs, we have overcome the bottleneck of producing SCCFs with well-arranged colloids. Not to mention, this aim was achieved using common polymers and within a relatively short timeframe.

Thanks to the Reviewer's comments, we have included some notes in the revised manuscript on page 19, as read: “Nevertheless, the decreased ordering in the top surface of the SCCF did not significantly impact its color quality. This is because both reflectance spectroscopy and visual inspection have a certain detection depth for evaluating the color quality of SCCF. The number of stacking layers (d') in the surface planes of the sample that can be detected by reflection spectroscopy depends on the refractive index contrast (Δn) and effective refractive index (\bar{n}) of the composites, which can be expressed as $d' = \bar{n}/\Delta n$. Therefore, for SCCF with a refractive index contrast of 0.061, the reflectance spectrometer can detect up to 26 layers of colloidal arrangement, corresponding to a detection depth of SCCF less than 6 μm . The SEM image in Fig. 4f shows that SCCF exhibited a highly long-range ordered structure with a depth of up to 32 μm . This indicates that although there are

some defects in the colloidal arrangement on the top surface of SCCF, the structural color with high brightness and saturation can still be observed by the naked eye and analyzed by reflectance spectroscopy. These results demonstrate that with the aim of achieving high-quality structural colors, SCCFs with well-arranged colloids were successfully prepared.” to explain this issue.

4. The authors provide a complete survey and comparison to support that their proposed approach has the advantages of low cost, high production efficiency, processing friendliness, simplicity, and scalability, do the authors have any data supporting?

Response: Thanks for the Reviewer for the valuable question. It should be noted that the comparison chart shown in Fig. 6c in the revised manuscript provides a qualitative comparison and analysis. To clarify the comparative basis of different strategies, we have compiled information on the variety of materials used, production efficiency, processing conditions, preparation steps, and production area for each strategy. We have listed the results in Table R1 below.

Table R1. Comparative analysis of strategic advantages

Strategies Comparative basis	Shearing force (supramolecular composites)	Shearing force (colloids with core-interlayer-shell structures)	Capillary force-induced colloidal assembly	Solvation force-induced colloidal assembly	Magnetic force-induced colloidal arrangement
Materials	Commercial polymers and commonly-used colloids	Colloids with core-interlayer-shell structures obtained by multi-step polymerization	Commonly-used colloids and commercial polymers or functionalized monomers	Commonly-used colloids and functionalized monomers	Magnetic colloids, e.g., Fe ₃ O ₄ -based colloids and functionalized monomers
Production efficiency	Seconds to minutes	Seconds to minutes	Hours to days	Hours	Seconds to minutes
Processing conditions	Room temperature processing	Generally above room temperature processing	Room temperature processing and sometimes UV curing	Additional UV curing	UV and magnetic fields
Preparation steps	Compositing colloid with polymer and shearing	Multi-step covalent grafting of polymers to colloids and shearing	Co-assembly of colloids with polymers or self-assembly of colloids and introduction of polymers	Co-assembly of colloids with polymer monomers, infiltration into the gap between two glass slides, and UV curing of monomers	Compositing colloid with polymer monomers, magnetic field-induced colloid assembly, and UV curing of monomers

Production area	Square meter	Square meter	Square meter	Square centimeter	Square centimeter
References	This work	[R1-R5]	[R6-R8]	[R9-R11]	[R12]

Furthermore, we also quantitatively calculated the material cost and production efficiency of fabricating SCCFs in our study using shearing supramolecular composites as a reference to further demonstrate the advantages of this strategy in terms of low cost and high production efficiency. Taking the example of SCCFs prepared by simply mixing commonly available PS-COOH colloids (or silica colloids) with a commercially available PEI, the cost of the PS-COOH and silica colloids is ~60 \$/kg and ~198 \$/kg, respectively. PEIs were purchased from Wuhan Lullaby Pharmaceutical Chemical Co., Ltd. at ~16 \$/kg or from Sigma Aldrich at ~290 \$/kg. As a result, the total material costs of the SCCF can be as low as ~\$0.004-0.015/m² (~\$0.027-0.038/m² for silica colloids). On the other hand, the production efficiency of the SCCF can reach 0.43 m²/min. Moreover, the SCCFs can be obtained at the square meter scale by simply physically mixing the colloids with the polymer and continuously winding them using roll-to-roll equipment without the need for complex chemical synthesis methods or high-end synthesis equipment, demonstrating the simplicity of the fabrication process for SCCFs. In addition, the supramolecular interactions in the supramolecular composite are dynamically reversible, and the processing modulus of the composite can be lowered by introducing a small amount of water, thus giving the SCCF a unique advantage in room temperature processing.

Thanks to the Reviewer's helpful question, to provide the basis for the comparison in Fig. 6c of the revised manuscript, we have added Table R1 in the revised Supplementary Information as Supplementary Table 2.

5. The application of the flexible SCCFs could be considered to explore.

Response: Thanks to the Reviewer's suggestion. This study aims to report a scalable and mild strategy for the preparation of SCCF via shearing supramolecular composites and to reveal the mechanism of shear-induced colloidal ordering, which lays the foundation for the construction and application of flexible SCCFs. In fact, we have conducted preliminary explorations in various applications of flexible SCCFs, including anti-counterfeiting, sensing, and coatings. The results hopefully will be reported separately in the near future. As a response, we have attached the SCCFs to phone cases and also

created stylish wristbands to showcase the potential application of flexible SCCFs in the field of decoration. The results are detailed in Figure R1 below. Besides, the unique angle-dependent structural color properties of SCCF have important applications in the anti-counterfeiting field.

Figure R1. Applications of flexible SCCF as decoration of phone cases (a) and stylish wristbands (b).

Thanks again for the Reviewer's suggestions, we have added the potential application of flexible SCCFs in the revised Supplementary Information as Supplementary Fig. 31.

6. Please add scale in Fig. 4a, Fig. 4d, and Fig. 1d for reference.

Response: Thanks for the helpful suggestion for further improvement. As suggested by the Reviewer, we have added scale bars in Fig. 4a, Fig. 4d, and Fig. 1d for reference in the revised manuscript. Additionally, we have double-checked the entire manuscript to correct similar issues.

7. The authors investigate the role of polymer molecular weight on shear-induced colloidal ordering, why are higher molecular weights of PEI polymer not chosen (more than 60 k).

Response: We appreciate the Reviewer for raising this thought-provoking question. In our manuscript, we overlooked the influence of higher molecular weight polymers on the shear-induced ordering mechanism. To rectify this omission, we have used PEI with a molecular weight of 2000 kDa to investigate its impact on the shear-induced colloidal ordering mechanism. The detailed experimental results are provided in Figure R2 below.

Figure R2. Reflection spectra of the SCCFs with PEI of molecular weight of 2000 kDa before and after shearing treatments. Insets: the corresponding photographs.

We found that as the PEI molecular weight increased to 2000 kDa, the reflection peak intensity of the SCCF after sufficient shearing decreased to 4%. This means that compared to the PEI with a molecular weight of 60 kDa, the shear-induced ordering mechanism of PEI with a molecular weight of 2000 kDa is no longer significant. This may be due to the fact that ultra-high molecular weight polymers are prone to excessive entanglement, which enhances the interaction between polymeric matrices, thus leading to energy dissipation due to excessive chain entanglement and ultimately making it difficult to achieve effective momentum transfer.

We appreciate the Reviewer's question, and we have added Figure R2 to the Supplementary Information as Supplementary Fig. 13 and added some notes in the revised manuscript to explain further the role of higher molecular weights of PEI on shear-induced colloidal ordering, as read, “It is noteworthy that the influence of polymer molecular weight on the shear-induced ordering effect does not show a strict positive correlation. When the PEI molecular weight increased from 60 to 2000 kDa, the peak intensity of the sufficient sheared SCCF decreased from 55% to 4% (Supplementary Fig. 13). This indicates that the shear-induced ordering effect significantly diminished with further increase in PEI molecular weight. This phenomenon may be attributed to the tendency of ultra-high molecular weight PEIs to undergo excessive entanglement, which enhances interactions between polymer matrices, increases energy dissipation, and ultimately hinders effective momentum transfer.” on pages 12-13.

Reviewer #2 (Remarks to the Author):

In the manuscript by Li et al., the authors reported on a strategy to produce large-scale structurally colored composite films by shearing supramolecular composites composed of polymers and colloids with supramolecular interactions. The authors effectively exploit the supramolecular connections between polymers and colloids, enabling the ordered arrangement of colloids through efficient momentum transfer. Furthermore, the dynamic nature of supramolecular interactions is harnessed to enhance the processability of the composite. The authors also provided a comprehensive elucidation of the mechanism behind shear-induced colloidal ordering and demonstrated the remarkable generality and versatility of this approach. This work provided an innovative approach to producing large-scale structurally colored composite films with attractive properties, which are challenging. Overall, the manuscript is well written, and I believe it will appeal to the broad readership of Nature Communications. I recommend publication after the following questions have been clarified.

Response: Thanks for the Reviewer's positive comments on our innovative approach to producing large-scale structurally colored composite films.

1. The authors elucidated that the shear-induced colloidal ordering in supramolecular composites is related to the molecular weight of PEI. Therefore, understanding the rheological properties of pure PEI is critical when constructing structurally colored composite films using shearing methods. It is recommended to add this information to the main text and discuss it further.

Response: We appreciate the Reviewer's valuable suggestion. As suggested, we have moved the rheological properties of pure PEI in Supplementary Fig. 9a to Fig. 2c and added relevant discussions in the revised manuscript on page 11, as read, "To elucidate the mechanism of shear-induced colloidal ordering in supramolecular composites, the viscoelastic properties of these PEIs and their corresponding composites were analyzed through rheological measurements, specifically the frequency dependence of the shear moduli (i.e., storage modulus (G') and loss modulus (G'')). It should be noted that for PEI₆₀₀, its viscosity was independent of frequency (Supplementary Fig. 10), implying that PEI₆₀₀ behaved like a Newtonian fluid with negligible elasticity⁵⁶. For PEI_{10k}, its G'' was proportional to ω^2 , and G' was proportional to ω (Fig. 2c), indicating that PEI_{10k} was a typical nonentangled polymer⁵⁶. In contrast, the G' and G'' of PEI_{60k} were highly dependent on frequency, implying that PEI_{60k} was a typical viscoelastic liquid caused by chain entanglement⁵⁶."

2. Can these samples be recycled as the composites are obtained based on supramolecular interactions? In addition, how is the stability of the structure or optical properties of structurally colored composite films?

Response: Thanks for the insightful questions. Due to the dynamic reversibility provided by supramolecular interactions, the SCCFs exhibited excellent recyclability. Specifically, we subjected the SCCFs to a series of cyclic treatments, including dissolution with water, lamination, hot pressing, and re-shearing. After that, the reflection wavelength and peak intensity of the SCCFs with different cycle numbers were measured. The detailed experimental results are provided in Figure R3 below. We found that the reflection wavelength and corresponding peak intensity of the SCCF remain almost unchanged with increasing cycle numbers. This indicates that the SCCFs possess good recyclability.

Figure R3. Reflection peak wavelength and peak intensity of SCCFs after sufficient shearing treatments. Insets: corresponding photographs of the initial and cyclic 6th SCCFs.

We admitted that supramolecular interactions can be affected by solvent or high-temperature treatments. However, under ambient storage conditions, SCCFs exhibited excellent optical stability. Specifically, we investigated the optical properties of SCCFs after 3 months of storage through optical imaging and reflectance spectroscopy. The results are shown in Figure R4 below. We compared the optical properties of SCCF after 3 months of storage with the initial SCCF to study the optical stability of SCCFs. We found that the reflection peak intensity and photographs of SCCF stored at room temperature for 3 months showed almost no change compared to the initial sample. This result

indicates that SCCF possesses outstanding optical stability.

Figure R4. Reflection spectrum and photographs (insets) of SCCF after 3 months of storage at room temperature at viewing angles of 0° and 60° (sample diameter: 2.2 cm)

We appreciate the Reviewer’s questions. To further highlight the advantages of supramolecular strategies, we have added Figure R3 and Figure R4 to the Supplementary Information as Supplementary Fig. 29 and Supplementary Fig. 30, and added some descriptions of recyclability and optical stability of SCCF in the revised manuscript, as read, “**Moreover, we found that SCCFs obtained from shearing supramolecular composites possessed stable optical properties at room temperature and can be cyclically reused with the assistance of a suitable solvent, which further enhances the practical applicability of SCCFs (Supplementary Fig. 29 and Supplementary Fig. 30).**” on page 23.

3. In Figure 4c, the authors compellingly demonstrate the superiority of the shearing supramolecular composite strategy by comparing the optical properties of structurally colored composite films obtained through different approaches, using reflection peak intensity and normalized FWHM as metrics. However, it is crucial to acknowledge that the optical quality of structurally colored composite films is intricately linked to the material's refractive index. Therefore, it is strongly recommended to include the refractive index of the polymer matrix and colloid, even if provided in the Supplementary Information (SI) section, to ensure a comprehensive evaluation of the optical characteristics.

Response: Thanks for the helpful suggestion. To provide a more accurate and comprehensive comparison of the optical properties of SCCFs, we have summarised the refractive index information

for the colloids and polymer matrices of the SCCFs shown in Fig. 4c of the revised manuscript in Table R2.

Table R2. Refractive indices of the matrix polymer and colloid of SCCFs

Reference	Refractive index of colloid	Refractive index of polymer matrix	Refractive index contrast
This work	1.590	1.529	0.061
Ref.19	1.450	~1.48	~-0.03
Ref.28	1.910	1.460	0.450
Ref.30	1.450	1.502	0.052
Ref.33	1.590	1.455	0.135
Ref.36	2.420	1.460	0.960
Ref.37	1.590	1.479	0.111
Ref.38	1.590	1.436	0.154
Ref.39	1.590	1.436	0.154
Ref.43	1.590	1.479	0.111

We appreciate the Review's suggestion and have added Table R2 to the Supplementary Information as Supplementary Table 1.

4. Characterization of Colloid Size and Distribution: The manuscript lacks information on the size and distribution of colloids.

Response: Thanks for the helpful suggestion for further improvement. We have performed SEM imaging and particle size distribution analysis for the colloids of different sizes mentioned in the manuscript. The detailed experimental results are provided in Figure R5 below.

Figure R5. SEM images and the corresponding histograms of the size distribution of colloids with average diameters of 177 nm (a, g), 182 nm (b, h), 188 nm (c, i), 192 nm (d, j), 203 nm (e, k), and 225 nm (f, i).

We appreciate the Review's suggestion and added Figure R5 in the Supplementary Information as Supplementary Fig.1.

5. Impact of Water Content (More than 20 wt%): The manuscript should address the potential impact of high water content (more than 20 wt%) on processability. The optimal upper limitation of water content is not clear.

Response: Thanks for the helpful suggestion for further improvement. We supplemented and compared the processing performance of supramolecular composites with water contents of 25 wt.% and 30 wt.%. The detailed experimental results are provided in Figure R6 below.

Figure R6. Reflection spectra of SCCFs with water contents of 25 wt.% (a) and 30 wt.% (b) after sufficient shearing treatment. Insets: corresponding photographs.

Through shearing the supramolecular composite with a water content of 25 wt.%, we found that the peak intensity of the corresponding SCCF only reached a maximum of 37%. Comparatively, the optical quality significantly decreased when shearing the supramolecular composite with a water content of 25 wt.% (Figure R6a). Furthermore, when the water content of the supramolecular composite increased to 30 wt.%, the corresponding peak intensity dropped to 4%, and the structural color became almost negligible (Figure R6b, inset). In summary, these results demonstrated that the optical properties of the SCCFs initially increased and then decreased as the water content of the supramolecular composite increased, with an optimal processing performance observed at a water content of 20 wt.%. This result is consistent with the influence of the volume fraction of colloids on the shear-induced colloidal ordering mechanism and can be explained as follows: as the water content increases from 0 wt.% to 20 wt.%, the modulus of the supramolecular composite decreased, facilitating the migration of colloids under shearing; as the water content increased to 30 wt.%, the elastic properties gradually deteriorated, impeding the transfer of colloidal momentum under shearing. Therefore, the optimal processing performance is ultimately achieved with a water content of 20 wt.% in the supramolecular composite.

We appreciate the Reviewer's suggestion. We have added Figure R6 to the Supplementary Information as Supplementary Fig. 20 and added relevant discussions in the revised manuscript to clarify the current optimal processing performance, as read: **“To meticulously examine the impact of the water content on the processability, a series of PEI_{60k}-PS₆₀ composites with different water contents were prepared. We found that for the PEI_{60k}-PS₆₀ composites with water content of 10 wt.%, 15 wt.%,**

and 20 wt.%, the reflection peak intensity after sufficient shearing can reach 60%, while for PEI_{60k}-PS₆₀ composites with water content as high as 25 wt.% and 30 wt.%, the reflection peak intensity significantly decreased to 37% and 4%, respectively (Supplementary Fig. 20). This may be due to the gradual deterioration of the elastic properties at high water content, which hinders the transfer of colloidal momentum during shearing. Based on these results, the processing performance of PEI_{60k}-PS₆₀ composites with water content of 10 wt.%, 15 wt.%, and 20 wt.% was further studied in detail.”
on page 20.

Reference

- R1. Zhao, Q. et al. Large-scale ordering of nanoparticles using viscoelastic shear processing. *Nat. Commun.* **7**, 11661 (2016).
- R2. Li, H., Wu, P., Zhao, G., Guo, J. & Wang, C. Fabrication of industrial-level polymer photonic crystal films at ambient temperature based on uniform core/shell colloidal particles. *J. Colloid Interface Sci.* **584**, 145-153 (2021).
- R3. Li, H. et al. Polychrome photonic crystal stickers with thermochromic switchable colors for anti-counterfeiting and information encryption. *Chem. Eng. J.* **426**, 130683 (2021).
- R4. Huang, H. et al. Gecko-inspired smart photonic crystal films with versatile color and brightness variation for smart windows. *Chemical Engineering Journal* **429**(2022).
- R5. Huang, H. et al. Butterfly-inspired tri-state photonic crystal composite film for multilevel information encryption and anti-counterfeiting. *Adv. Mater.* **35**, 2211117 (2023).
- R6. He, Y., Liu, L., Fu, Q. & Ge, J. Precise assembly of highly crystalline colloidal photonic crystals inside the polyester yarns: a spray coating synthesis for breathable and durable fabrics with saturated structural colors. *Adv. Funct. Mater.* **32**, 2200330 (2022).
- R7. Li, M. et al. Structure-tunable construction of colloidal photonic composites via kinetically controlled supramolecular crosslinking. *Macromolecules* **55**, 8345-8354 (2022).
- R8. Tan, H. et al. Metallosupramolecular photonic elastomers with self-healing capability and angle-independent color. *Adv. Mater.* **31**, e1805496 (2019).
- R9. Wu, Y., Wang, Y., Zhang, S. & Wu, S. Artificial chameleon skin with super-sensitive thermal and mechanochromic response. *ACS Nano* **15**, 15720-15729 (2021).
- R10. Lee, G. H. et al. Chameleon-inspired mechanochromic photonic films composed of non-close-packed colloidal arrays. *ACS Nano* **11**, 11350-11357 (2017).
- R11. Yang, Q. et al. Large-scale production of high-quality elastic structural color films based on hydrogen bond and colloidal charge co-driven silica microsphere self-assembly. *Chem. Eng. J.* **455**, 140591 (2023).
- R12. Xie, Y. et al. Bistable and reconfigurable photonic crystals-electroactive shape memory polymer nanocomposite for ink-free rewritable paper. *Adv. Funct. Mater.* **28**, 1802430 (2018).

REVIEWERS' COMMENTS

Reviewer #1 (Remarks to the Author):

After revision, the authors did quite a few works to improve the quality of paper. I would recommend this manuscript for this format to be accepted and published in NC.

Reviewer #2 (Remarks to the Author):

In my opinion, the authors have made proper revisions on the manuscript, which has address my concerns. Therefore, I would like to recommend the publication of the paper in the Nat. Commun.